# WNK1 Kinase Stimulates Angiogenesis to Promote Tumor Growth and Metastasis

**DOI:** 10.3390/cancers12030575

**Published:** 2020-03-02

**Authors:** Zong-Lin Sie, Ruei-Yang Li, Bonifasius Putera Sampurna, Po-Jui Hsu, Shu-Chen Liu, Horng-Dar Wang, Chou-Long Huang, Chiou-Hwa Yuh

**Affiliations:** 1Institute of Molecular and Genomic Medicine, National Health Research Institutes, Zhunan, Miaoli 35053, Taiwan; zonlins@gmail.com (Z.-L.S.); 074041@nhri.edu.tw (R.-Y.L.); boni_bt123@nhri.edu.tw (B.P.S.); ray60115@yahoo.com.tw (P.-J.H.); 2Institute of Biotechnology, National Tsing-Hua University, Hsinchu 30013, Taiwan; hdwang@life.nthu.edu.tw; 3Department of Biomedical Sciences and Engineering, National Central University, Jhongli Dist., Taoyuan 32001, Taiwan; jennyliu66@gmail.com; 4Division of Nephrology, Department of Internal Medicine, University of Iowa Carver College of Medicine, Iowa, IA 52242, USA; 5Institute of Bioinformatics and Structural Biology, National Tsing-Hua University, Hsinchu 30013, Taiwan; 6Department of Biological Science & Technology, National Chiao Tung University, Hsinchu 30010, Taiwan; 7Ph.D. Program in Environmental and Occupational Medicine, Kaohsiung Medical University, Kaohsiung 80708, Taiwan

**Keywords:** with-no-lysine (K) kinase I, tumor-induced angiogenesis, zebrafish, hepatocellular carcinoma, colorectal cancer

## Abstract

With-no-lysine (K)-1 (WNK1) is the founding member of family of four protein kinases with atypical placement of catalytic lysine that play important roles in regulating epithelial ion transport. Gain-of-function mutations of WNK1 and WNK4 cause a mendelian hypertension and hyperkalemic disease. WNK1 is ubiquitously expressed and essential for embryonic angiogenesis in mice. Increasing evidence indicates the role of WNK kinases in tumorigenesis at least partly by stimulating tumor cell proliferation. Here, we show that human hepatoma cells xenotransplanted into zebrafish produced high levels of vascular endothelial growth factor (VEGF) and WNK1, and induced expression of zebrafish wnk1. Knockdown of *wnk1* in zebrafish decreased tumor-induced ectopic vessel formation and inhibited tumor proliferation. Inhibition of WNK1 or its downstream kinases OSR1 (oxidative stress responsive kinase 1)/SPAK (Ste20-related proline alanine rich kinase) using chemical inhibitors decreased ectopic vessel formation as well as proliferation of xenotransplanted hepatoma cells. The effect of WNK and OSR1 inhibitors is greater than that achieved by inhibitor of VEGF signaling cascade. These inhibitors also effectively inhibited tumorigenesis in two separate transgenic zebrafish models of intestinal and hepatocellular carcinomas. Endothelial-specific overexpression of *wnk1* enhanced tumorigenesis in transgenic carcinogenic fish, supporting endothelial cell-autonomous effect of WNK1 in tumor promotion. Thus, WNK1 can promote tumorigenesis by multiple effects that include stimulating tumor angiogenesis. Inhibition of WNK1 may be a potent anti-cancer therapy.

## 1. Introduction

WNK (with-no-lysine [K]) kinases are a family of serine/threonine protein kinases with atypical kinase domain. There are four *WNK* genes in human [1,2]. Gain-of-function of WNK1 and WNK4 caused by gene mutation leads to an autosomal-dominant disease pseudohypoaldosteronism type 2 (PHA2) featured by hypertension, hyperkalemia and hyperchloremic metabolic acidosis [2,3]. These findings lead to the understanding of the role of WNK kinases in regulating epithelial ion transport through downstream ste20-related proline/alanine-rich kinase (SPAK) and related oxidative-stress response kinase-1 (OSR1) as well as some kinase-independent actions of WNKs [4,5,6,7,8,9,10]. WNKs are also expressed outside epithelial tissues and involved in many other functions [11,12,13]. Among these, WNK1 is ubiquitously expressed and we have found that it is essential for embryonic angiogenesis in mouse [14,15]. We have also found that zebrafish has two *wnk1* genes, *wnk1a* and *1b*, and each is required in embryonic angiogenesis during development [16]. The function of wnk1 in embryonic angiogenesis in zebrafish is downstream of vascular endothelial growth factor (VEGF) signaling [16].

WNK1 has been reported to be involved in many aspects of carcinogenesis. For example, WNK1 is required for activation of extracellular signal-regulated kinases 5 (ERK5) by epidermal growth factor (EGF) and Mitogen-activated Protein/ERK Kinase Kinases (MEKK2/3) pathway [17], and Wnt/β-catenin signaling [18]. WNK1 is linked to oncogenic phosphoinositide-3-kinase (PI3K)-AKT pathways, transforming growth factor beta (TGF-β) and Nuclear factor Kappa B (NF-κB) pathways [11]. Together, these reports indicate that WNK1 plays an important role in tumor cell growth and remodeling of extracellular matrix for tumor invasion. However, the role of WNK1 in tumor-induced angiogenesis remains unknown.

Nutrient supply by blood vessels is essential for the growth of solid tumor beyond a certain size. Many cancers produce high levels of VEGF to induce formation of new blood vessels. The process known as tumor-induced angiogenesis is a hallmark of cancer development that has long been employed as an anti-cancer therapeutic target. Given that WNK1 plays a role in embryonic angiogenesis and may have multiple tumor-promoting effects, in this study we investigate the role of WNK1 in tumor-induced angiogenesis and the overall effect of WNK1 inhibition on tumor growth and metastasis.

## 2. Results

### 2.1. Identify a Hepatoma Cell Line (Hep3B) with High Angiogenic Activity and Establish Stable Hep3B_Lifeact-RFP Cells for Xenotransplantation Study

Tumor xenotransplantation is an effective way to study tumor-induced ectopic blood vessel formation. Zebrafish (*Danio rerio*) is a favorable model for studying angiogenesis owing to features including conserved genome to human, short life cycle, transparent embryos, and *Tg (fli1:EGFR)* transgenic fish that marks the whole vasculature [19,20,21,22]. We designed experiments to examine the role of WNK1 in tumor-induced angiogenesis using xenotransplantation in zebrafish.

We first examined the expression levels of VEGFA from several tumor cell lines for selecting tumor cells with the highest angiogenic activity for xenotransplantation. We found that all hepatoma cells examined have higher levels of VEGFA expression than colorectal cancer (CRC) cells (Figure 1A). Among these, Hep3B human hepatoma cells exhibited the highest levels of VEGFA.

To allow tracing of transplanted tumors, we generated a stable Hep3B cell line in which actin filament is labelled by red fluorescence protein by lentiviral transfection of plasmid encoding Lifeact-RFP (Hep3B_Lifeact-RFP). Fluorescence-activated cell sorting (FACS) was used to select cells with high Lifeact-RFP expression (Figure 1B,C). Stable expression of Lifeact-RFP did not affect the ability of Hep3B cells to produce VEGFA (Figure 1D) and nor did it on the cell size or granularity (Figure 1E). We have used flow cytometry and found the stable expression of Lifeact-RFP did not affect the cell size or granularity (Appendix A).

### 2.2. Xenotransplanted Hep3B_Lifeact-RFP Cells Induce Angiogenesis in Zebrafish Embryos

To determine whether Hep3B_Lifeact-RFP hepatoma cells can induce angiogenesis, they were injected into 2 day-post-fertilization (dpf) *Tg (fli1:EGFP)* embryos, and interaction between red fluorescence-labeled tumor cells and green fluorescent blood vessels was observed under microscopy. As shown, Hep3B_Lifeact-RFP cells induced ectopic vessel branching from subintestinal vein (SIV) at 1 day-post-injection (dpi) (Figure 1F). At 2–4 dpi, new blood vessels sprouted and tumor cells invaded into blood vessels and migrated to distant locations. The results indicate that xenotransplanted Hep3B tumor induces angiogenesis in zebrafish embryos.

### 2.3. Knockdown wnk1 Reduces Tumor-Induced Angiogenesis and Prevents Tumor Cell Proliferation in Zebrafish Embryos

To investigate the role of wnk1 in tumor-induced angiogenesis, *Tg (fli1:EGFP)* embryos at 1-cell stage were injected with morpholino (MO) antisense oligonucleotides, followed by injection of Hep3B_Lifeact-RFP cells at 2 dpf, and analyzed between 0 and 3 dpi. Compared to controls, *wnk1a* and *wnk1b* morphants exhibited lower percentage of embryos with ectopic vessels formation (Figure 2A,B). The effects of knockdown *wnk1a* and *wnk1b* are similar to knockdown of *flk1* which encodes VEGF receptor type 2 (VEGFR2). We quantified the angiogenic response by measuring the number and length of ectopic vessels after morpholino knockdown wnk1. Supporting the notion that wnk1 is important for tumor angiogenesis, the average number (Figure 2C) and length (Figure 2D) of ectopic vessels in *wnk1* morphants were much lower compared to controls. We further examined the effect of *wnk1* knockdown on tumor cell proliferation. We quantified tumor-associated RFP fluorescence and reported the results as the percentage embryos exhibiting decreases (green zone), unchanged (blue zone) or increases (red zone) in fluorescence in 1 dpi versus 0 dpi (Figure 2E) and as well as the percentage change in fluorescence in 1 dpi versus 0 dpi (Figure 2F). In both metrics, *wnk1a* knockdown reduced proliferation of Hep3B cells relative to controls, however, the effect of *wnk1b* knockdown versus controls did not reach statistical significance. This phenomena is consistent with our previous study in embryonic angiogenesis [16], whereas *wnk1a* knockdown has more significant reduction embryonic angiogenesis than *wnk1b*. Overall, the results indicate that knockdown of *wnk1a* gene in zebrafish reduces tumor-induced angiogenesis and tumor cell proliferation, and knockdown of *wnk1b* is less effective in both angiogenesis and tumor cell proliferation reduction. As expected from the fact that *wnk1* is essential for embryonic angiogenesis during development [16], *wnk1* knockdown (like *flk1* knockdown) resulted in higher embryos mortality (Figure 2G).

### 2.4. Inhibition of WNK Kinase Signaling Cascade by Small Molecule Inhibitors Is Anti-Angiogenic

To circumvent embryo lethality and to evaluate potential clinical application of WNK inhibition in cancer therapy, we investigated effects of chemical inhibitors of WNK signaling cascade on angiogenesis. WNK463 is the first orally bioavailable pan-WNK kinase inhibitor targeting to the WNK kinase domains. It is highly selective for WNK kinases due to their unique kinase domain structure [23]. WNK1 regulates embryonic angiogenesis by activating downstream kinase OSR1 [15]. Closantel is an FDA approved broad-spectrum antiparasitic that targets to the highly conserved C-terminal domain of SPAK and OSR1 to prevent their activation by WNK kinases [24]

We first tested the effects of WNK463 and Closantel on embryonic angiogenesis. *Tg (fli1:EGFP)* embryos were immersed with WNK signaling inhibitors from 1 dpf, and observed for the effects on the length of the intersegmental vein (ISV) for subsequent 24 h. Compare to negative (1% DMSO) and positive control (VEGFR inhibitor PTK787) (Appendix A), WNK463 dose-dependently inhibited the development of ISV (Appendix A). The half-maximal inhibitory concentration (IC_50_) for WNK463 was calculated at 35.04 ± 3.81 μM (Appendix A). Supporting the notion that OSR1 acting downstream of WNK1 in embryonic angiogenesis, Closantel inhibited the development of ISV (Appendix A). The IC_50_ for Closantel was calculated at 2.04 ± 0.33 μM (Appendix A).

### 2.5. WNK1 and OSR1/SPAK Inhibitors Exhibit Stronger Anti-Proliferation Activity in Xenotransplanted Tumor than VEGFR Inhibitor

Unlike embryonic angiogenesis studies that measure the development of ISV at ~30 h post-fertilization, xenotransplantation studies are conducted on embryos over many days post-fertilization when significant death occurs from *wnk1* knockdown (Figure 2G). Thus, we first investigated and determined that the optimal concentrations of WNK463 and Closantel for xenotransplantation assay were 2.5 and 0.15 μM, respectively (Appendix A).

To examine the effect of WNK1 inhibitor on xenotransplanted tumor, Hep3B_Lifeact-RFP cells were injected into the yolk of zebrafish embryos at 2 dpf, and embryos were immersed with drugs one day later and observed for additional two days. Embryos treated with PTK787, WNK463 or Closantel had reduced tumor size compared to the DMSO control (Figure 3A–C). The mean fluorescence area and mean fluorescence intensity of tumor-associated fluorescence (shown as percentage changes 3 dpi versus 1 dpi) treated with WNK463 or Closantel were much lower compared to the DMSO control (Figure 3D–E). The results indicate that WNK1 pathway inhibitors exhibit stronger antitumor activity than the VEGFR inhibitor PTK787 in the xenotransplantation assay, although this result might be simply due to the different efficacy of these inhibitors, another possibility is that WNK kinases might also promote tumor cell proliferation autonomously within the cancer cells, we will examine the role of WNK kinases using endothelial-specific wnk1 overexpression transgenic fish (Section 2.8).

### 2.6. Oral Gavage of WNK463 and Closantel Reduce Tumorigenesis in Adult Zebrafish Colorectal Cancer Model

Transgenic fish that overexpresses ribose-5-phosphate isomerase A (RPIA) under the control of an intestinal fatty acid-binding protein (*ifabp*) promoter (*Tg(ifabp:RPIA; myl7:EGFP*) developed cancer in the gut at ~3 months of age [25]. RPIA converts ribulose-5-phosphate to ribose-5-phosphate, a substrate for nucleotide biosynthesis. We used the RPIA transgenic fish as a model to examine the effects of WNK1 signaling inhibitors on colorectal cancer (CRC).

Three-month-old transgenic RPIA fish were administered WNK463 or Closantel by oral gavage for one month, and intestinal specimens were collected for molecular and pathological examination. All fish were survived, only one wild-type (WT) fish that received the DMSO vehicle control died during the one-month experimental period (Appendix A) indicating minimal toxicity of WNK463 and Closantel. We analyzed the following molecular markers by QPCR: *RPIA*, *cyclin D1* (*ccnd1)*, *wnk1a* and *wnk1b*, *flk1*, and *matrix metalloproteinase 9*, and *mmp9*. The *ccnd1* is required for cell cycle and cell proliferation in CRC, while *mmp9* is involved in the breakdown of extracellular matrix, which is important for tumor neovascularization and metastasis. The expression levels of all genes examined were upregulated in transgenic RPIA fish, and compared to vehicle (DMSO) treatment, treatment with WNK463 or Closantel significantly reversed upregulation of *ccnd1*, *mmp9*, *wnk1a* and *wnk1b* in RPIA transgenic fish (Figure 4A–F). The reduction of *ccnd1* and *mmp9* supports the notion that WNK463 and Closantel inhibit colon tumorigenesis, metastasis, and tumor angiogenesis. Tumor cells produce VEGF, which plays a role in transcriptional regulation of wnk1 expression [16]. This fact may explain the finding that inhibitors decrease *wnk1a* and *wnk1b* expression.

Tissue from multiple regions of the intestinal tract were collected from control fish, treated and untreated RPIA transgenic fish, and subjected to histopathological examination after hematoxylin and eosin (H&E) staining, and to analysis of proliferating cell nuclear antigen (PCNA) by immunohistochemical staining (IHC). As shown in representative H&E-stained sections, features indicative of tumorigenesis including decreased nuclear-to-cytoplasmic ratio, nuclear condensation, and atypia (Figure 4G), and cellular hyperplasia, dysplasia and neoplasia (Figure 4H) were observed in RPIA transgenic versus control fish. Compared to transgenic fish treated with vehicle control, treatment with WNK463 or Closantel ameliorated tumorigenesis in transgenic fish (Figure 4G,H). In IHC staining of PCNA, signal was markedly increased in transgenic RPIA fish versus control fish (Figure 5A,B). Treatment with WNK463 or Closantel significantly reduced PCNA signal compared to the DMSO control. Thus, WNK1 signaling cascade inhibitors are effective in preventing CRC tumorigenesis in adult fish.

### 2.7. Oral Gavage of WNK1 Signaling Inhibitors Reduces Tumorigenesis in Adult Zebrafish Hepatocellular Carcinoma Model

We further examined the efficacy of inhibition of WNK1 pathway in hepatocellular carcinoma (HCC) using an adult transgenic fish that overexpresses src tyrosine kinase in the liver driven by the hepatitis B protein x (HBx). The transgenic fish, *Tg (fabp10a:HBx-mCherry, src; my17:EGFP)* normally develops HCC at 11 months, but tumorigenesis occurs at ~5-months after two-month overfeeding to induce obesity [26]. We first validated that the HBx src transgenic fish developed obesity-induced HCC in 5-months. WNK463, Closantel or vehicle was then orally administered to the transgenic fish for one month, and liver specimens were collected for molecular and pathological examination. As reported previously, HBx, src transgenic fish gained more weight than WT fish (Appendix A), probably due to liver hyperplasia [26]. Body width and length were not different. One month’s feeding with WNK463 significantly decreased body weight in transgenic fish compared to vehicle-treated transgenic fish (Appendix A). The mean body weight of the Closantel treated group showed trend to be lower than the vehicle control, although not statistically significant (Appendix A). Treatment with WNK463 or Closantel did not affect the survival (Appendix A).

Liver tissues were harvested from WT and transgenic fish after one month oral gavage of vehicle or inhibitors. The expression of lipogenic factor (carbohydrate-responsive element-binding protein, *chrebp1*), fibrosis maker gene (heparanase, *hpse*), *ccnd1*, *wnk1a*, *wnk1b*, *flk1/vegfr2* and *mmp9* were measured by QPCR. All genes except *hpse* were upregulated in transgenic hepatoma fish (Appendix A). Inhibition of WNK signaling had no significant effect on the expression of *chrebp1,* a marker of steatosis (Appendix A). Steatosis can occur from overfeeding besides being a part of tumorigenesis. Compared to the vehicle-treated group, treatment with WNK463 or Closantel significantly reduced the overexpression of *ccnd1* and *mmp9* in the transgenic fish (Appendix A). The expression of *wnk1a*, *wnk1b* and *flk1* were unchanged by WNK463 or Closantel treatment.

Histopathological changes were examined by H&E staining. Treatment with WNK463 reduced dysplasia compared to DMSO control, and Closantel reduced steatosis compared to DMSO control (Appendix A). Thus, WNK1 inhibitors prevent HCC tumorigenesis in zebrafish.

### 2.8. Endothelial-Specific Wnk1 Overexpression Enhances Hepatocarcinogenesis in HBx, src, (p53-) Transgenic Fish

WNK signaling cascade inhibitors caused relatively greater inhibition on tumor cell proliferation than *wnk1* knockdown (see Figure 3D versus Figure 2F). To understand the basis of the difference, we uncovered that xenotransplanted hepatoma cells produced WNK1 (Figure 6A). Interestingly, endogenous wnk1a and wnk1b production were also induced on day 2 and 3 after tumor injection (Figure 6B,C). Thus, multiple sources or targets of WNK1 exist.

To support the notion that WNK1 indeed stimulates angiogenesis to contribute to tumor growth, we generated transgenic fish *Tg (fli1:wnk1a; my17:EGFP)* that drives *wnk1a* expression in endothelium via *fli1* promoter and crossed with *Tg (fabp10a:HBx-mCherry, src; my17:EGFP; p53-)* transgenic fish which carries mutations in *p53* tumor suppressor, such that hepatoma develops with an accelerated rate [26]. The expression of *HBx*, *ccne1 (Cyclin E1)*, *ccnd1*, *src*, *wnk1a*, and *flk1* was analyzed by QPCR from two separate transgenic lines. The *HBx* transgene was overexpressed in all transgenic lines (Figure 7A). The expression of *wnk1a* in *HBx, src, (p53-)* transgenic fish was at the baseline level as WT, but markedly increased in the wnk1a-overexpressing lines, confirming the overexpression (Figure 7B). The expression of the *ccne1* and *src* oncogenes was slightly increased in *HBx, src, (p53-)* transgenic fish, and further increased by the endothelial *wnk1a* overexpression (Figure 7C,D). Expression of *ccnd1* and *flk1* were not consistently or significantly affected in *HBx, src, (p53-)* transgenic fish nor by endothelial *wnk1a* overexpression (Figure 7E,F). Histopathological examination revealed that endothelial overexpression of *wnk1a* increased hepatic tumorigenesis in H&E-stained liver sections (Figure 7G–J,O) and proliferation marker PCNA analyzed by IHC (Figure 7K–N,P).

We used WNK463 treatment to further support the role of endothelial *wnk1a* overexpression in enhancing hepatic tumorigenesis. WNK463 feeding decreased wnk1a-stimulated hepatic tumorigenesis (Figure 8A–F,M). The effect of WNK463 on PCNA staining was statistically significant for TG line 1, but not for line 2 (Figure 8G–L,N).

## 3. Discussion

Hypertension and hyperkalemia in pseudohypoaldosteronism type-2 (PHA2) patients with *WNK1* and *WNK4* mutations lead to the understanding of roles of WNK kinases in ion homeostasis. Increasing evidence indicates that WNK signaling pathway is also involved in cancer formation and progression [11]. WNKs intersect with many known oncogenic pathways including PI3K/AKT, ERKs, TGF-β, to regulate tumor cell proliferation, migration, and metastasis [27]. In the study, we show that WNK1 signaling pathway plays an important role in tumor-induced angiogenesis. Inhibition of WNK1 signaling cascade either by gene knockdown or oral admini stration of inhibitors decreases tumor-induced angiogenesis, proliferation and growth of tumor. These effects are observed in xenotransplanted hepatoma as well as transgene-induced colorectal cancer and hepatoma in zebrafish.

Literature suggests multiple potential mechanisms for WNK1-OSR1/SPAK signaling cascade to regulate endothelial cells for formation of blood vessels. Transmembrane Na^+^, K^+^, and Cl^-^ ion fluxes and calcium signaling affect migration of many cell types including endothelial cells [28,29]. WNK1 is a positive regulator of expression of transcription factors Slug, MMP2, ZEB1, which control endothelial cell migration [30]. Interestingly, in human umbilical vein endothelial cells, SPAK is required for endothelial cell proliferation whereas OSR1 is required for migration [30]. Our findings from using WNK1 and OSR1/SPAK inhibitors support the notion that OSR1/SPAK acts downstream of WNK1 for tumor angiogenesis and tumor growth.

Anti-angiogenesis by targeting VEGF is an important component of anticancer therapeutic regimens [31]. However, for some tumors the benefits of VEGF signaling inhibition may be transitory, followed by a restoration of tumor growth and progression. The resistance may be due to variance of cell types, tumor microenvironment, compensatory and/or alternate angiogenic signaling. Thus, combination therapy aiming at multiple targets is desirable.

In this study, WNK463 and Closantel exhibit stronger anti-tumor activity than VEGFR inhibitor PTK787. These results could be explained on the basis of a different efficacy of these inhibitors. This finding might also imply that WNK kinases have other tumor-promoting effects including direct proliferative actions on tumors [11,32]. Similar to WNK1 downstream target-SPAK functions both downstream and also upstream of NF-κB signaling [33,34] forming a positive feedback loop to promote tumorigenesis. WNK1 might also play multiple roles in tumorigenesis through cross-talk with multiple signal pathways. We also find that tumors produce WNK1 and that endogenous zebrafish wnk1 is induced by xenotransplanted tumor. We have previously reported that VEGF regulates *wnk1* expression [16], which likely explains the finding of WNK1 production in tumor as well as in tumor-transplanted fish. Among multiple potential sources and targets of WNK1, we further demonstrate that endothelial cell-autonomous effect of WNK1 on tumors does occur.

Binding of VEGF to its receptors in endothelial cells activates multitudes of intracellular signaling cascades including PI3K-AKT, phospholipase C-γ (PLCγ), src, p38 mitogen-activated protein kinase (MAPK) that promote endothelial cell survival, proliferation, migration, vascular permeability, and artery-vein fate specification [35]. WNK1 is a positive regulator of canonical Wnt/β-catenin signaling in cancer cell line [18]. Wnt/β-catenin pathway promotes cell proliferation, cell survival, and expression of angiogenic factor in endothelial cells [36]. Therefore, WNK1 may affect tumor cell proliferation and angiogenesis by activating Wnt/β-catenin signaling. Our recent report on embryonic angiogenesis in zebrafish reveals an additional mechanism for VEGF to promote angiogenesis by PI3K-AKT-mediated phosphorylation of wnk1 at an amino-terminal threonine [25]. WNK1-SPAK/OSR1 regulates proliferation, invasion and gene expression in endothelial cells [30]. We have found that OSR1 acts downstream of WNK1 in embryonic angiogenesis [15]. The current finding that Closantel inhibits tumor angiogenesis provides further support for the notion that VEGF-WNK1-OSR1 cascade is important for endothelial cell function in angiogenesis.

## 4. Materials and Methods

### 4.1. Ethics Statement

All zebrafish experiments were approved by the Institutional Animal Care and Use Committee (IACUC) of the National Health Research Institutes of Taiwan (NHRI) (protocol number: NHRI-IACUC-108035-A) and were in accordance with International Association for the Study of Pain guidelines.

### 4.2. Zebrafish Maintenance and Transgenic Zebrafish Lines

The zebrafish embryos, larvae, and adult fish were maintained at 28 ℃ at the Zebrafish Core Facility of NHRI. Adult fish were maintained with continuous flow under an automated 14:10 h light–dark cycle. Tg *(ifabp:RPIA; myl7:EGFP)* contains an intestinal fatty acid-binding protein (*ifabp*) promoter to drive the expression of human ribose-5-phosphate isomerase A (RPIA). RPIA is involved in pentose phosphate pathway (PPP) to convert ribulose-5-phosphate to ribose-5-phosphate that is the critical step from the oxidative phase into the non-oxidative phase in the PPP. Ribose-5-phosphate is used for nucleotide biosynthesis. Tg *(ifabp:RPIA; myl7: EGFP)* transgenic fish develop tumorigenesis in the gut of adult fish [25]. Tg *(fabp10a: HBx-mCherry, src; myl7: EGFP)* contains a fatty acid-binding protein-10a (*fabp10a*) promoter to drive the expression of Hepatitis B virus x protein (HBx) and *src* oncogene. Tg *(fabp10a: src, HBx-mCherry; myl7: EGFP)* transgenic fish develop hepatocellular carcinoma (HCC) at 11-months old [29]. The tumorigenesis process in zebrafish can be sped up to 5-months by overfeeding for 2-months to force diet induced-obesity [26]. Tg *(fliI: wnk1a; myl7: EGFP)* fish carry *wnk1a* gene under the control of *fli1* endothelial-specific promoter, this transgenic fish were crossed with Tg *(fabp10a: HBx-mCherry, src; myl7: EGFP, p53-)* to generate endothelial *wnk1a* overexpression in HBx, src, p53- transgenic fish.

### 4.3. Embryos Collection

One day prior fertilization, the adult male and female zebrafish were placed into mating tanks with a divider to prevent mating. The next morning, the divider was removed to start the mating process and embryos were collected 1 h later. After discarding dead and unfertilized embryos, embryos were transferred into 90 mm petri dish with E3 medium (5 mM NaCl, 0.17 mM KCl, 0.33 mM CaCl_2_ and 0.33 mM MgSO_4_) and incubated at 28 °C.

### 4.4. Microinjection

Morpholino anti-sense oligonucleotide (MO), mRNAs or DNAs were prepared in PBS with 0.05% (w/v) phenol red. Embryos were injected at the one-cell stage to the site between animal pole and vertebral pole via glass capillary using a Nanoject II™ Nanoliter injector (Drummond Scientific). For MO injection, 2.5–5 ng in 4.6 nL were microinjected into 1-cell stage embryos. For DNA and RNA co-injection, 50–150 pg in 2.3 nL were microinjected into 1-cell stage embryos. After injection, embryos were washed with E3 and incubated at 28 °C.

### 4.5. Cell Culture

Human cancer cell lines were cultured in complete Dulbecco’s modified Eagle’s medium (C-DMEM) (Gibco, Carlsbard, CA, USA) supplemented with 10% fetal bovine serum (FBS), 1% penicillin and 1% streptomycin. All the cell lines were maintained at 37 °C incubator with 5% CO_2_.

### 4.6. Establishment of Hep3B_Lifeact-RFP Stable Line

Hep3B_Lifeact-RFP cells were transfected with rLVUbi-Lifeact-TagRFP lentiviral vector (ibidi, Martinsried, Planegg, Germany) and screened for RFP+ cells with 0.5 mg/mL puromycin (Gibco, USA). Fluorescence-activated cell sorting (FACS) was used to isolate RFP+ cells with BD Influx cell sorter (Becton Dickinson, Franklin Lakes, NJ, USA) by the Cell Sort Core Lab (CSCL) in the NHRI.

### 4.7. Morpholino Anti-Sense Oligonucleotide

The morpholinos were custom-made by Gene Tools (USA), dissolved to 1 mM in sterilized ddH_2_O and stored at −80 °C. Before MO microinjection, the injection mixture was heated to 65 °C for 10 min to avoid aggregation. The sequence of MOs are: Control MO: 5′-CCTCTTACCTCAGTTACAATTTATA-3′; flk-1 MO: 5′-GTCTGTTAAAATAACGTCCCGAAATG-3′; wnk1a MO: 5′-TCCACCAAGTGGGAGCGTGAAGTTAG-3′; wnk1b MO: 5′-TGCGTAAATTTCCTGCTCTTGCTT-3′. The dosage of morpholino used in this study was based on previous titration results [16]. Then, 2.5 ng control MO, 2.5 ng flk1 MO, 5 ng wnk1a MO, and 5ng wnk1b MO were microinjected.

### 4.8. Xenotransplantation of Hepatoma in Zebrafish

One-cell stage Tg *(fli1:EGFP)* embryos were microinjected with morpholino. At 2 dpf, Tg *(fli1:EGFP)* embryos were dechlorinated with pronase (Sigma-Aldrich Inc., St. Louis, MO, USA) and anaesthetized. Trypsinized Hep3B_Lifeact-RFP cells were collected, counted, washed, and resuspended in 1× PBS. We injected 4.6 nL of single-cell suspensions which contained 200 Hep3B_Lifeact-RFP cells into each embryo at the middle of yolk sac. After xenograft, embryos were washed with PTU/E3 and incubated in a RI-80 low temperature incubator. Temperature was gradually increased from 28 to 37 °C in 2 days, and maintained at 37 °C until the end of the experiment.

For observation the angiogenesis and proliferation of Hep3B_Lifeact-RFP cells, 0–3 dpi embryos were anaesthetized, placed on a 1% agarose plate for microscopy under a SZX10 stereo microscope equipped with DP71 digital camera and U-LH100HG fluorescence light source with U-RFL-T power supply (Olympus, Shinjuku, Tokyo, Japan). Red and green fluorescent images were obtained. Images of Hep3B_Lifeact-RFP cells from 0-3 day-post-injection (dpi) were quantified for fluorescent intensity as a readout of tumor cell proliferation activity. Length and number of ectopic vessels from subintestinal vein (SIV) were determined as indices of angiogenic activity. All the images were quantified using Image J software.

### 4.9. Resource of Compounds

Closantel (Cat. No.: HY-17596) and WNK463 (Cat. No.: HY-100626) were purchased from MedChem Express (Monmouth Junction, NJ, USA). Vatalanib dihydrochloride (PTK787, Catalog No.S1101) was purchased from Selleckchem Crop. (Houston, TX, USA).

### 4.10. Confocal Microscopy

For living confocal imaging, embryos were anaesthetized and mounted in 1% PTU/E3-low melt agarose (Zymest) on 3.5mm glass bottom petri dish to avoid movement. Fluorescent images (for EGFP and RFP) were collected using Leica TCS SP5 confocal microscope.

### 4.11. Oral Gavage

Zebrafish were anaesthetized, placed on a wet sponge to prevent movement, and fed with 5 μL drug solution using microliter syringes (HAMILTON, Salt Lake City, UT, USA) and FTP-22-25 plastic feeding tubes (INSTECH, Plymouth Meeting, PA, USA). Fish were put back to fresh water immediately after oral gavage for recovery from anaesthetization. Dosage of WNK463 and Closantel were 510 ng/fish and 425 ng/fish, respectively. Fish were weighted, length measured before and after oral gavage.

### 4.12. Tissue preparation, Histological, and Immunochemical Analysis

Liver and intestinal tissues were harvested. The intestinal tract was divided into three parts including intestinal bulb (IB), middle-intestine (MI) and posterior-intestine (PI). About 2/3 of the tissues were frozen in liquid nitrogen instantly after dissection and then store at −80 °C for later RNA isolation and further QPCR analysis. The remaining parts were fixed in a 10% formalin solution, embedded in paraffin, sectioned at 5 μm thickness, mounted on poly-L-lysine-coated slides, and stained with hematoxylin-eosin (HE) or performed immunohistochemistry (IHC) staining as previously described by us [19,37].

### 4.13. RNA Isolation, Real Time Quantitative Polymerase Chain Reaction (QPCR)

Total RNA from cells and tissue were extracted with NucleoSpin^®^ RNA kit (MACHEREY-NAGEL, Bethlelm, PA USA). About 30 mg of cells or tissue were collected and stored at -80 °C immediately after collection. For RNA isolation, tissues were thawed, homogenized, and lysates were processed for RNA isolation using NucleoSpin^®^ RNA column per manufacturer‘s protocol. Isolated RNA samples were determined for concentration using a NanoDrop ND-1000 UV-Vis Spectrophotometer and stored at −80 °C. Complementary DNA (cDNA) was synthesized using iScript™ cDNA synthesis kit (BioRad, Hercules, CA, USA). cDNA was diluted 100-fold for QPCR analysis using SYBR Green (Thermo Fisher Scientific, Waltham, MA, USA). Primers are listed in Table 1. All experiments were performed in triplicate and at least three independent samples were used for QPCR.

### 4.14. Statistical Analysis

All the statistical analysis of the qPCR results was performed using unpaired Student’s t-test, Figure 1A was analyzed by one-way ANOVA; histopathological data were analyzed by Chi-square test. A *p*-value smaller than 0.05 was considered to be statistically significant and is shown as: *: 0.01 < *p* ≤ 0.05, **: 0.001 < *p* ≤ 0.01 and ***: 0.0001 < *p* ≤ 0.001; ****: *p* ≤ 0.0001.

## 5. Conclusions

Overall, functional diversity of WNK1 provides multiple positive feedback loops for amplification of tumor growth (Figure 9). Thus, WNK1 signaling cascade may be a multi-purpose target for combination cancer therapy.

## Figures and Tables

**Figure 1 cancers-12-00575-f001:**
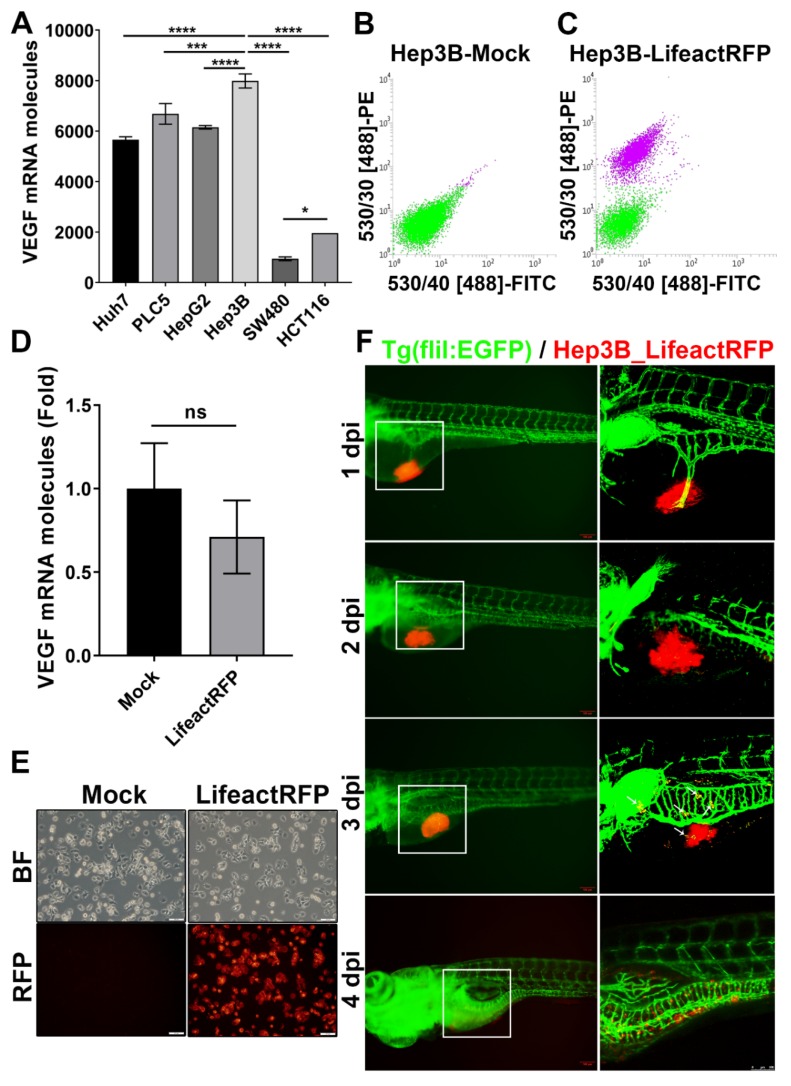
Establishment of stable RFP-expressing cancer cell line for tumor xenotransplantation study. (**A**) VEGF (VEGFA) mRNA expression in human hepatocellular carcinoma (Huh7, PLC5, HepG2, Hep3B) and colorectal cancer (SW480, HCT116) cell lines. The statistical significance was analyzed by one-way ANOVA. ns: non-significant, *p* > 0.05; *: 0.01 < *p* ≤ 0.05; ***: *p* ≤ 0.001; ****: *p* ≤ 0.0001; (**B,C**) FASC of un-transfected (B) and rLVUbi-Lifeact-TagRFP transfected Hep3B cells (C). The purple cell pools were sorted to create a stable Hep3B_Lifeact-RFP cell line. (**D**) mRNA expression of VEGFA in Hep3B and Hep3B_Lifeact-RFP stable line. ns, *p* > 0.05; unpaired Student’s t-test. (**E**) Representative bright field and fluorescent images of Hep3B and Hep3B_Lifeact-RFP stable line. Scale bar = 50 μm. (**F**) Representative images of Hep3B_Lifeact-RFP injected embryos from 1~4 day-post-injection (dpi) taken by fluorescent microscope (left) and confocal microscope (right). White arrow indicates metastatic cells at 3 dpi. Scale bar = 100 μm.

**Figure 2 cancers-12-00575-f002:**
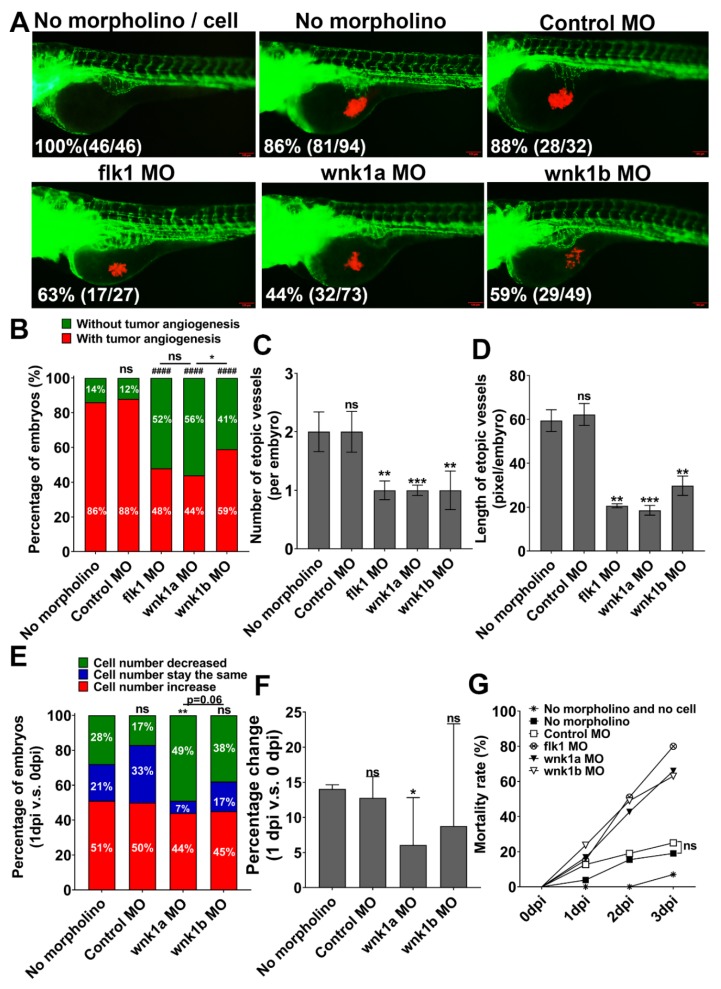
Effect of wnk1 knockdown on tumor-induced angiogenesis and tumor cell proliferation in zebrafish embryos. (**A**) Representative images of control embryos and embryos injected with Hep3B_Lifeact-RFP ± indicated morpholino. Images were at 2 dpi. Percentage of live embryos (number of live embryos/total embryos) for each experimental group was shown. Scale bar = 100 μm. (**B**) Percentage of embryos with (red zone) and without angiogenic response (green zone). Total number of embryos was 24. The statistical significance was analyzed by Chi-square test. ####: *p* ≤ 0.0001; ns: non-significant compared to the no morpholino control. ns: non-significant compared flk1 MO and wnk1a MO; *: 0.01 < *p* ≤ 0.05 compared wnk1a MO and wnk1b MO. (**C,D**) The average number of ectopic vessels (C) and length of ectopic vessels (D) in different morphants. The statistical significance was analyzed by unpaired Student’s t-test. **: 0.001 < *p* ≤ 0.01; ***: 0.0001 < *p* ≤ 0.001 compared to the no morpholino control. (**E**) Distribution of embryos with decreases (green zone), unchanged (blue) and increases in tumor-associated fluorescence in 1 dpi relative to 0 dpi. Changes of more than 5% on either direction were defined as decrease or increase. Total number of embryos was 24. The statistical significance was analyzed by Chi-square test. **: 0.001 < *p* ≤ 0.01; ns: non-significant compared to the no morpholino control. (**F**) Percentage changes of the tumor-associated fluorescent intensity in 1 dpi from 0 dpi. ns: non-significant, *p* > 0.05; *: 0.01 < *p* ≤ 0.05 compared to the no morpholino control. (**G**) Mortality rate of different morphants.

**Figure 3 cancers-12-00575-f003:**
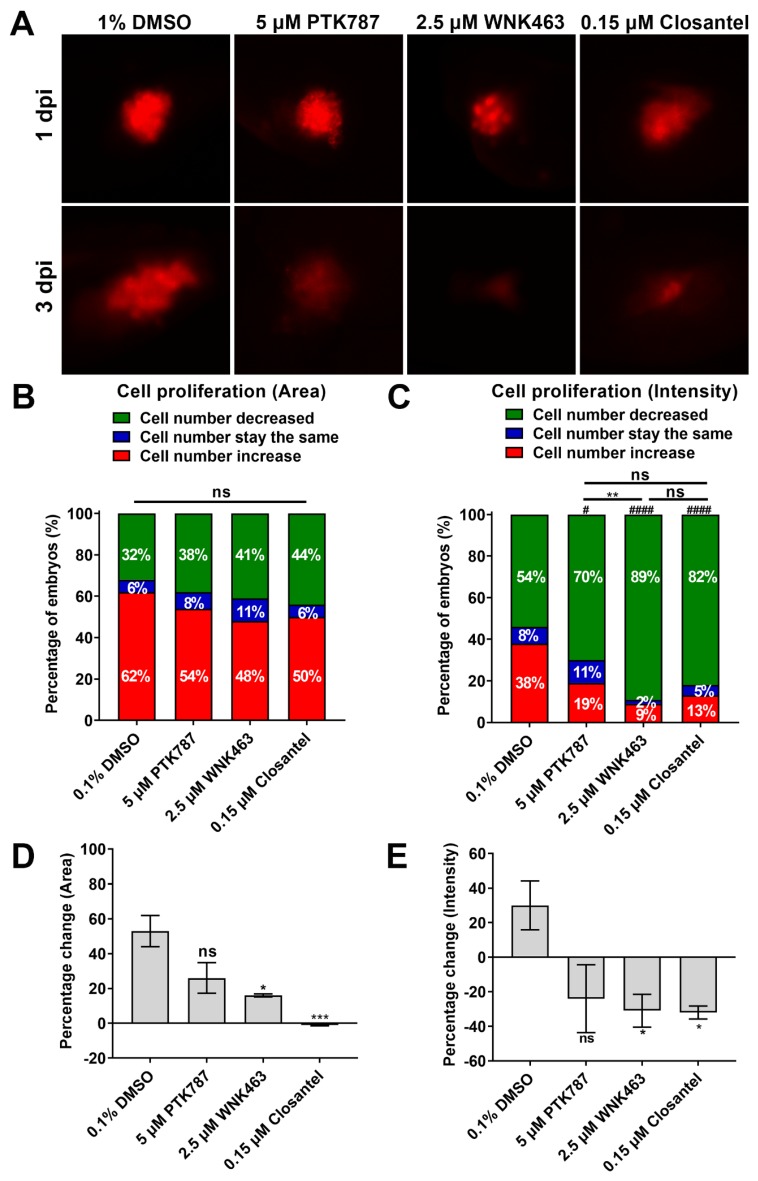
Effect of VEGF receptor inhibitor (PTK787), WNK inhibitor (WNK463) and OSR1/SPAK inhibitor (Closantel) on proliferation of xenotransplanted hepatoma cells. (**A**) Representative images of embryos before and after treated with DMSO (vehicle), PTK787, WNK463 or Closantel. Fish were immersed with drugs for 2 days beginning at 1 dpi (day postinjection) of hepatoma cells. (**B,C**) Distribution of embryos with decreases (green zone), unchanged (blue) and increases in tumor-associated fluorescence area (**B**) and intensity (**C**) after drugs. Total number of embryos was 24. The statistical significance was analyzed by Chi-square test. ns: non-significant, *p* > 0.05 comparison between different inhibitors. #: 0.01 < *p* ≤ 0.05; ####: *p* ≤ 0.0001 compared to the 0.1%DMSO control. **: 0.001 < *p* ≤ 0.01 compared PTK787 and WNK463. (**D,E**) Percentage changes of average fluorescent area (**D**) and average fluorescent intensity (**E**) at 3 dpi relative to 1 dpi. The statistical significance was analyzed by unpaired Student’s t-test. ns: non-significant, *p* > 0.05; *: 0.01 < *p* ≤ 0.05; ***: 0.0001 < *p* ≤ 0.001 compared to the 0.1% DMSO control.

**Figure 4 cancers-12-00575-f004:**
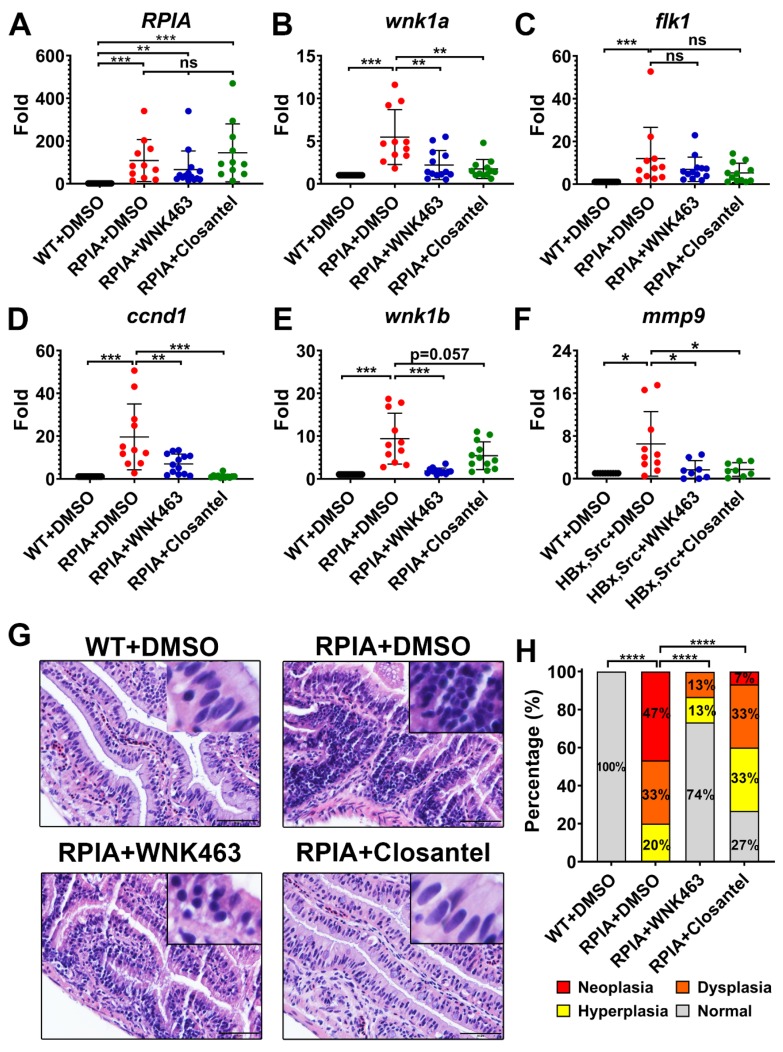
Effect of WNK1 pathway inhibitors on transgene-activated colorectal cancer (CRC) in zebrafish. (**A**–**F**) *Tg (ifabp:RPIA; myl7:EGFP)* adult fish were treated by oral gavage with DMSO (*n* = 15), WNK463 (*n* = 15) or Closantel (*n* = 15) and compared with wildtype fish treated with DMSO (*n* = 14). QPCR analysis was performed for zebrafish transgene ribose 5-phosphate isomerase A (*RPIA*), CRC promoting gene cyclin D1 (*ccnd1*), zebrafish WNK1 orthologue (*wnk1a* and *wnk1b*), VEGF receptor-2 (*flk1/vegfr2*), and angiogenesis/metastasis marker gene matrix metallopeptidase 9 (*mmp9*). The statistical significance was analyzed by unpaired Student’s t-test. ns: non-significant, *p* > 0.05; *: 0.01 < *p* ≤ 0.05; **: 0.001 < *p* ≤ 0.01; ***: 0.0001 < *p* ≤ 0.001; ****: *p* ≤ 0.0001. (**G**) Representative H&E-stained sections of intestinal tract of RPIA transgenic fish treated with or without inhibitors. Scale bar = 20 μm. (**H**) Distribution of histopathological findings from tissue sections as shown in panel G. The statistical significance was analyzed by Chi-square test. ****: *p* ≤ 0.0001.

**Figure 5 cancers-12-00575-f005:**
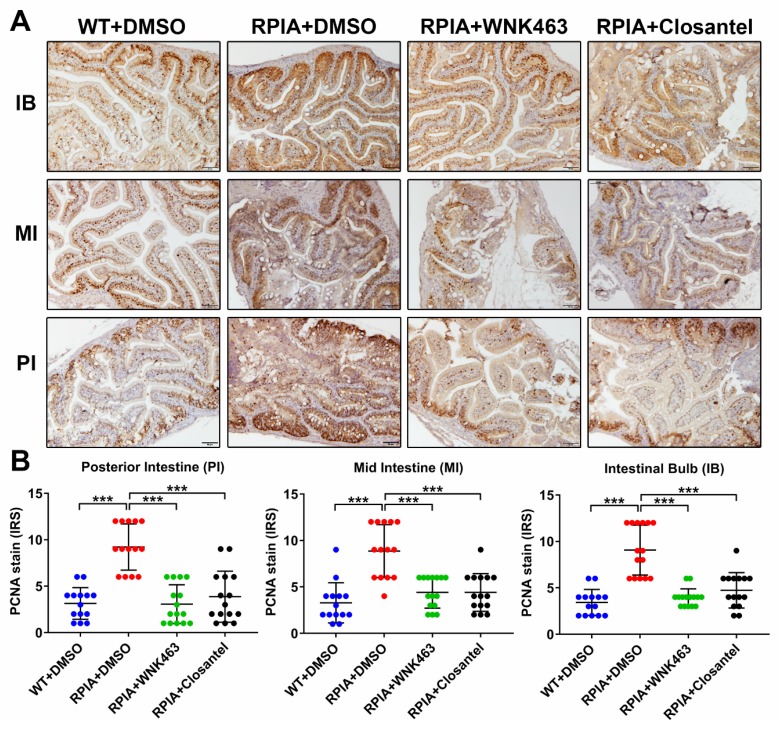
Effect of WNK1 pathway inhibitors on proliferating cell nuclear antigen (PCNA) expression in colorectal cancer. (**A**) Representative images and (**B**) immunoreactive score (IRS) of proliferating cell nuclear antigen (PCNA) staining on regions of intestine (IB: intestinal blub; MI: mid intestine; PI: posterior intestine) from WT and *Tg (ifabp:RPIA; myl7:EGFP)* fish treated with DMSO, WNK463 and Closantel. Scale bar = 50 μm. The statistical significance was analyzed by unpaired Student’s t-test. ***: *p* ≤ 0.001.

**Figure 6 cancers-12-00575-f006:**
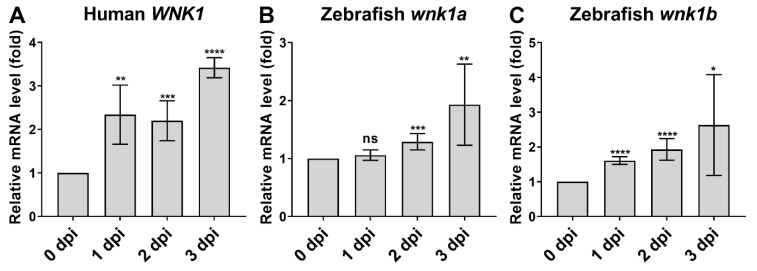
Xenotransplanted tumors produce WNK1 and induce endogenous WNK1 in zebrafish. (**A**) QPCR analysis of human WNK1 expression in xenotransplanted hepatoma at 0–3 dpi. (**B,C**) QPCR analysis of endogenous wnk1a and wnk1b in zebrafish 0–3 dpi hepatoma xenotransplantation. Statistical significance relative to 0 dpi. The statistical significance was analyzed by unpaired Student’s t-test. ns: non-significant, *p* > 0.05; *: 0.01 < *p* ≤ 0.05; **: 0.001 < *p* ≤ 0.01; ***: 0.0001 < *p* ≤ 0.001; ****: *p* ≤ 0.0001.

**Figure 7 cancers-12-00575-f007:**
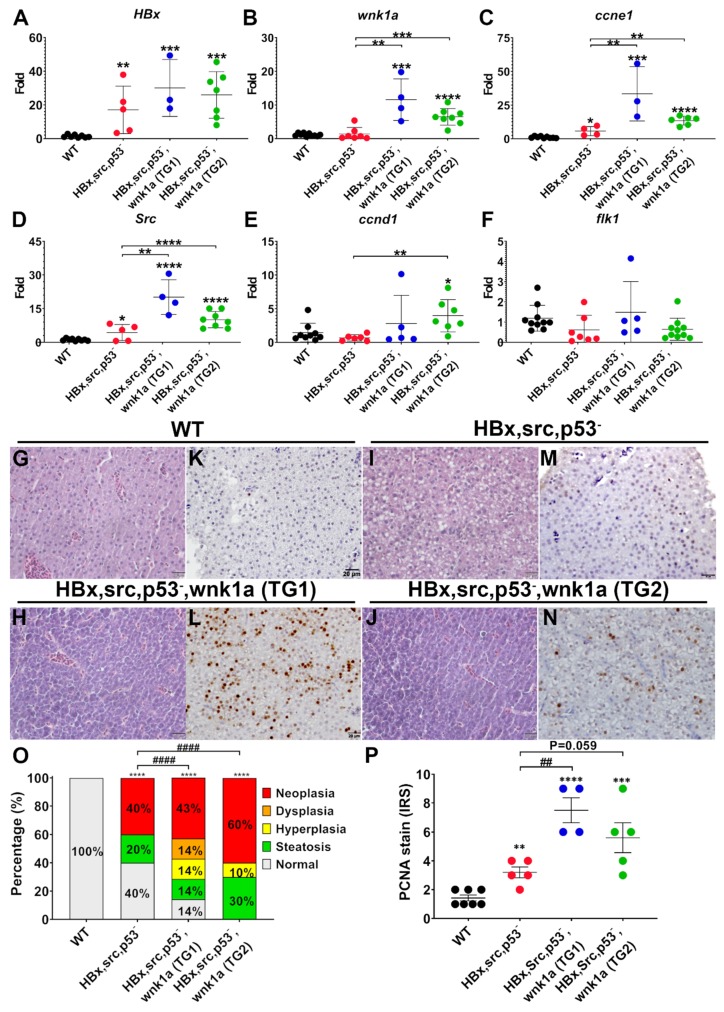
Endothelial cell-specific wnk1a expression promote tumorigenesis in HCC-transgenic fish. (**A**–**F**) QPCR analysis of genes in WT, HBx, src(p53-) fish, and HBx, src(p53-) fish overexpressing wnk1a in endothelium. Two separate transgenic lines, TG1 and TG2, were studied. (**G**–**J**) Representative images of H&E-stained liver sections. Scale bar = 20 μm. (**K–N**) Representative images of immunohistochemical staining of liver sections for proliferating cell nuclear antigen (PCNA). Scale bar = 20 μm. The statistical significance was analyzed by unpaired Student’s t-test. **: 0.001 < *p* ≤ 0.01; ***: 0.0001 < *p* ≤ 0.001; ****: *p* ≤ 0.0001 compared to WT control. (**O**) Distribution of histopathological findings of H&E-stained liver sections as shown in panels (G–J). The statistical significance was analyzed by Chi-square test. ****: *p* ≤ 0.0001 compared to WT control, ####: *p* ≤0.0001 comparison between transgenic fish. (**P**) Average immunoreactive score (IRS) of proliferating cell nuclear antigen (PCNA) staining as shown in panels K-N. **: 0.001 < *p* ≤ 0.01; ***: 0.0001 < *p* ≤ 0.001; ****: *p* ≤ 0.0001 compared to WT control. ##: 0.001 < *p* ≤ 0.01 comparison between transgenic fish.

**Figure 8 cancers-12-00575-f008:**
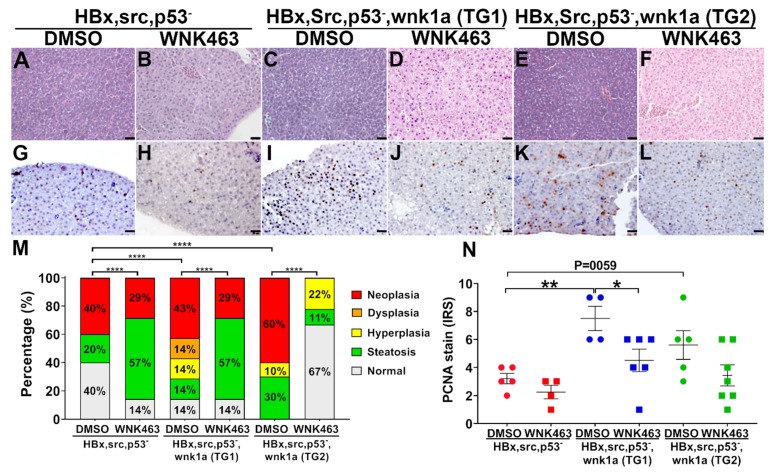
Effect of WNK1 pathway inhibitors on HCC in transgenic fish overexpressing endothelial cell-specific wnk1a. (**A**–**F**) Representative images of H&E-stained liver sections from HBx, src(p53-) fish and two lines (TG1 and TG2) of HBx, src(p53-) transgenic fish overexpressing wnk1a in endothelia. Scale bar = 20 μm. (**G–L**) Representative images of PCNA staining of liver sections from fish as above. (**M**) Distribution of histopathological diagnosis of H&E-sections as shown in panels (A–F). The statistical significance was analyzed by Chi-square test. ****: *p* ≤0.0001. (**N**) Average immunoreactive score (IRS) of proliferating cell nuclear antigen (PCNA) staining as shown in panels (G–L). The statistical significance was analyzed by unpaired Student’s t-test. *: 0.01< *p* ≤0.05; **: 0.001< *p* ≤0.01.

**Figure 9 cancers-12-00575-f009:**
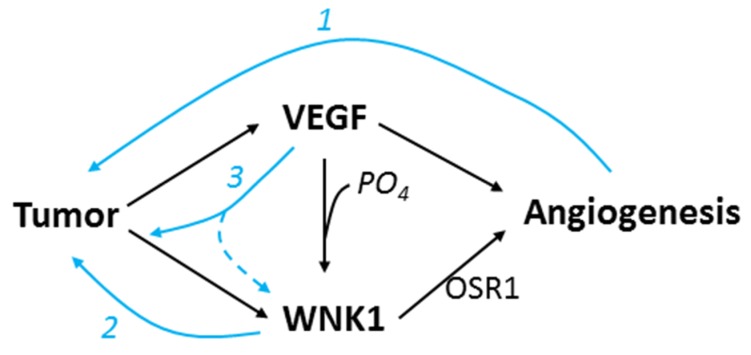
Model for WNK1 and VEGF interaction in angiogenesis and tumor growth. Tumor produces VEGF and WNK1, and both act to stimulate endothelial cells to form new blood vessels. WNK1 stimulates angiogenesis through downstream kinase OSR1. VEGF-stimulated angiogenesis is at least partly through PI3K/AKT-induced phosphorylation of WNK1. Three positive feedback loops participate in amplification of tumor growth: (1) angiogenesis providing nutrient for tumor to grow, (2) direct proliferative effect of WNK1 on tumor cells, and (3) expression of WNK1 from tumor or other tissues (dotted line) stimulated by VEGF.

**Table 1 cancers-12-00575-t001:** The primer information for qPCR analysis in human cell line and zebrafish.

Gene Name	Primer Name	Sequence (5′ to 3′)
*VEGFA*	*Q-hVEGFA-F*	GGGCAGAATCATCACGAAGT
	*Q-hVEGFA-R*	TGGTGATGTTGGACTCCTCA
*WNK1*	*Q-hWNK1-F*	GAGCGTCATCTGTGACTCCA
	*Q-hWNK1-R*	CGGTCTTTGCTGGTACTGCT
*EGFP* *Standard* *Curve*	*Q-EGFP-N1-F*	ACGTAAACGGCCACAAGTTC
	*Q-EGFP-N1-R*	AAGTCGTGCTGCTTCATGTG
	*pEGFP-N1-F*	CCTGATTCTGTGGATAACCGTAT
	*pEGFP-N1-R*	TGCATTCTAGTTGTGGTTTGTCC
*RPIA*	*Q-RPIA-F*	CATGCTGTGCAGCGAATAGC
	*Q-RPIA-R*	TGGCGGGCCTGGAAGGAAGT
*ccnd1*	*Q-ccnd1-F*	TTGCCTCTCATCCCAGAACCT
	*Q-ccnd1-R*	CCTGACACGATCGCAGACAGT
*wnk1a*	*Q-wnk1a-F*	TCGAGATAGGACGTGGCTCT
	*Q-wnk1a-R*	TCAAGGATGATTCCCAGGAG
*wnk1b*	*Q-wnk1b-F*	CCGGGTCAGCTGTGCCCAAG
	*Q-wnk1b-R*	TGGCCCAGGGGTTGTGAGGT
*mmp9*	*Q-mmp9-F*	GGTGATTGACGACGCCTTTG
	*Q-mmp9-R*	GAAATGGGCGTCTCCCTGAA
*flk-1*	*Q-flk1-F*	TCTTCACTCTTCACGTGCTTTTTAG
	*Q-flk1-R*	GAAGGTGTGTATCTCCATCAGGAA
*hpse*	*Q-hpse-F*	GCTCTGGTTTGGAGCTCATC
	*Q-hpse-R*	GAAATCCCGACCAAGTTGAA
*chrebp1*	*Q-chrebp-F*	GGAGATGGACTCGCTCTTTG
	*Q-chrebp-R*	GCAGAGGCTCAAAAGTGTCC
*actin*	*Q-actin-F*	CTCCATCATGAAGTGCGACGT
	*Q-actin-R*	CAGACGGAGTATTTGCGCTCA

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
