# Peer review of "WNK1 Kinase Stimulates Angiogenesis to Promote Tumor Growth and Metastasis"

_cancers, 2020, doi:10.3390/cancers12030575_

Round 1

Reviewer 1 Report

  1. Please avoid using abbreviation word in abstract section, ex OSR1
  2. Authors mentioned on line 88 that they quantified the number and length of ectopic vessel, but results was not found in figure?
  3. Where is the white arrow that authors mentioned in figure legend 1?
  4. It will be easier for reader to understand the figure if author provide information directly on figure, for example figure 3B and C. Reader can not recognize the difference between figure 3B and C directly through figure.
  5. Authors didn’t provide statistical analysis on these figures 1a, 2b, 2e, 3b, 3c.
  6. The total number of embryos of each figure is unclear and didn’t describe in manuscript, for example on 2b, 2e, 3b, 3c?
  7. The enlarge H&E images of figure 4g are very blurry, please provide more clear version.
  8. Seems PCNA is an important factor that WNK1 regulate tumor proliferation. Is it possible to involve the role of PCNA in your summarizing figure?
  9. Both WNK1a and WNK1b were induced by tumor; therefore, what is the role between WNK1a and WNK1b? The treatment target that author mention should be WNK1a or WNK1b.

Author Response

Comments and Suggestions for Authors

  1. Please avoid using abbreviation word in abstract section, ex OSR1

Ans: Thanks for the comments, we should write the full name instead of the abbreviation in the abstract.

  1. Authors mentioned on line 88 that they quantified the number and length of ectopic vessel, but results was not found in figure?

Ans: Thanks for the comments. This sentence is meant for the Fig 2C,D, so I moved it to line 111. The result of the number and length of ectopic vessel after morpholino knockdown were shown in Fig2C,D.

  1. Where is the white arrow that authors mentioned in figure legend 1?

Ans: Sorry for the missing arrows, we have updated the Fig 1F by adding arrows and box to indicate the enlarge area.

  1. It will be easier for reader to understand the figure if author provide information directly on figure, for example figure 3B and C. Reader can not recognize the difference between figure 3B and C directly through figure.

Ans: Thank you for your constructive suggestion. We should provide the information (red, green and blue indicated increased, decreased, and same cell number) on the top of figure.

  1. Authors didn’t provide statistical analysis on these figures 1a, 2b, 2e, 3b, 3c.

Ans. Thanks for the comments. We provide statistical analysis for the figures, Fig 1A was analyzed by one-way ANOVA; others were analyzed by Chi-square test.

  1. The total number of embryos of each figure is unclear and didn’t describe in manuscript, for example on 2b, 2e, 3b, 3c?

Ans. Thanks for the comments. We have added the description n=24 for each treatment in the figure legend, line 130 for 2b, line 135 for 2e, line 179 for 3b and 3c. All data are from at least 3 independent experimental replicates in the manuscript.

  1. The enlarge H&E images of figure 4g are very blurry, please provide more clear version.

Ans. Thank you for your constructive suggestion. We have provided more clear images for the enlarge HE stain for Fig 4g (line 214).

  1. Seems PCNA is an important factor that WNK1 regulate tumor proliferation. Is it possible to involve the role of PCNA in your summarizing figure?

Ans. Proliferating cell nuclear antigen (PCNA) is a DNA clamp that acts as a processivity factor for DNA polymerase δ in eukaryotic cells and is essential for replication, and we used PCNA as cell proliferation marker, it may not directly regulated by wnk1, so we didn’t include PCNA in our summarizing figure.

  1. Both WNK1a and WNK1b were induced by tumor; therefore, what is the role between WNK1a and WNK1b? The treatment target that author mention should be WNK1a or WNK1b.

Ans. Zebrafish often have duplicate copies of a gene that is present as a single copy in human orthologue. In this case, human WNK1 orthologue has duplicated copies in zebrafish: wnk1a and wnk1b, both were induced by tumor, and are inhibited by the drugs.

Reviewer 2 Report

In the present manuscript  Sie et al describe that inhibition of WNK1 signaling cascade either by gene knockdown or oral administration of inhibitors decreases tumor-induced angiogenesis proliferation and growth of tumor. They are observed these effects in xenotransplanted hepatoma as well as transgene-induced colorectal cancer and hepatoma in zebrafish. The manuscript is interesting but needs minor improvements.

Some minor points:

Line 77: The authors present representative bright field and fluorescent images of Hep3B and Hep3B_Lifeact-RFP stable line (Figure 1E) and comment that stable expression of Lifeact-RFP did not affect the cell size or granularity. Microscopic images are not sufficient to evaluate cell size or granularity but analysis by flow cytometry are more suitable. Line 87: The authors wrote “We quantified the angiogenic response by measuring the number and length of ectopic vessels at 1 dpi (see Figure 1F, top right panel)” but in the manuscript is not present. Furthermore the authors should show a negative control without xenotransplanted Hep3B tumor cells. Line 99: The authors wrote “White arrow indicates metastatic cells at 3 dpi.” but white arrow are not visible in the figure 1F. Line 122: the definition “Percentage of live embryos (# dead out of total #)” is unclear. Line 151: The authors determined that the optimal concentrations of WNK463 xenotransplantation assay is 2.5 μM (supplementary texts and associated Figure S2), but in Figure S2 there are only experiments with a concentration of WNK463 greater than or equal to 7.5 μM. line170: “Percentage changes of average fluorescent area (D) and average fluorescent intensity at ?? dpi relative to 1 dpi”.

Author Response

Comments and Suggestions for Authors

In the present manuscript Sie et al describe that inhibition of WNK1 signaling cascade either by gene knockdown or oral administration of inhibitors decreases tumor-induced angiogenesis proliferation and growth of tumor. They are observed these effects in xenotransplanted hepatoma as well as transgene-induced colorectal cancer and hepatoma in zebrafish. The manuscript is interesting but needs minor improvements.

Some minor points:

Line 77: The authors present representative bright field and fluorescent images of Hep3B and Hep3B_Lifeact-RFP stable line (Figure 1E) and comment that stable expression of Lifeact-RFP did not affect the cell size or granularity. Microscopic images are not sufficient to evaluate cell size or granularity but analysis by flow cytometry are more suitable.

Ans: Thank you for your constructive suggestion. We have used flow cytometry and found the stable expression of Lifeact-RFP did not affect the cell size or granularity. The figure was provided as Figure S1.

Line 87: The authors wrote “We quantified the angiogenic response by measuring the number and length of ectopic vessels at 1 dpi (see Figure 1F, top right panel)” but in the manuscript is not present. Furthermore the authors should show a negative control without xenotransplanted Hep3B tumor cells.

Ans: Sorry for the confusion. This sentence is meant for the Fig 2C,D, so I moved it to line 111. The result of the number and length of ectopic vessel after morpholino knockdown were shown in Fig2C,D. The negative control are shown in Fig2A, top left panel.

Line 99: The authors wrote “White arrow indicates metastatic cells at 3 dpi.” but white arrow are not visible in the figure 1F.

Ans: Sorry for the missing arrows, we have updated the Fig 1F by adding arrows and box to indicate the enlarge area.

Line 122: the definition “Percentage of live embryos (# dead out of total #)” is unclear.

Ans: Thank you for the suggestion, we have re-write as “percentage of live embryos (number of live embryos/total embryos) for each experimental group were shown.

Line 151: The authors determined that the optimal concentrations of WNK463 xenotransplantation assay is 2.5 μM (supplementary texts and associated Figure S2), but in Figure S2 there are only experiments with a concentration of WNK463 greater than or equal to 7.5 μM.

Ans: Thanks for the comments, in the first batch of xenotransplantation assay, we had treated tumor implanted embryos with several dosages; because tumor implanted embryos had a higher mortality rate (67 and 63% when treated with 5uM and 7.5uM WNK463), we therefore decreased the dosage to 2.5uM (29% mortality rate). We have added the data to the figure S2 (C, F).

line170: “Percentage changes of average fluorescent area (D) and average fluorescent intensity at ?? dpi relative to 1 dpi”.

Ans: Sorry for the confusion, we should re-write as “Percentage changes of average fluorescent area (D) and average fluorescent intensity at 3 dpi relative to 1 dpi”

Reviewer 3 Report

Sie et al. present an interesting and well written paper that suggests WNK1 as a promising pharmacological target to inhibit cancer angiogenesisa and cancer growth. All teh experiments are in zebrafish.

The following revisions should be made before the pubblication of the paper:

Introduction: This part must be deepened. In particular litterature data on the role of WNK1 in cancer should be described. Page 4 line114-115. Only one out of two wnk1 knockdown reach statistical significance on tumor cell proliferation. A third wnk1 knockdown should be employed to support the conclusion that "the knkcdown of wnk1 gene in zebrafish reduces...tumor cell proliferation". Authors compare the effect of Wnk1 and OSR1 inhibitors with VEGFR inhibitor reaching the conclusion that the involvement of Wnk1 in angiogenesis is more important than VEGFR. This is NOT correct. These results can be explained on the basis of a different efficacy of these inhibitors. This part should be changed and discussed also in conclusion section.

Author Response

Comments and Suggestions for Authors

Sie et al. present an interesting and well written paper that suggests WNK1 as a promising pharmacological target to inhibit cancer angiogenesis and cancer growth. All the experiments are in zebrafish.

The following revisions should be made before the publication of the paper:

Introduction: This part must be deepened. In particular literature data on the role of WNK1 in cancer should be described.

Ans: Thanks for the comments, we have added the literature on the role of WNK1 in hepatocellular carcinoma, breast cancer, retinoblastoma and cervical carcinoma in the introduction. Avoid duplication, we focus on the general role of WNK1 in introduction and the role in cancer in discussion. 

Page 4 line114-115. Only one out of two wnk1 knockdown reach statistical significance on tumor cell proliferation. A third wnk1 knockdown should be employed to support the conclusion that "the knockdown of wnk1 gene in zebrafish reduces...tumor cell proliferation".

Ans: Thanks for the comments, both wnk1a and wnk1b knockdown reduces tumor cell proliferation, but due to the big variation in wnk1b knockdown, there is no significance.

Authors compare the effect of Wnk1 and OSR1 inhibitors with VEGFR inhibitor reaching the conclusion that the involvement of Wnk1 in angiogenesis is more important than VEGFR. This is NOT correct. These results can be explained on the basis of a different efficacy of these inhibitors. This part should be changed and discussed also in conclusion section.

Ans: Thanks for the constructive comments, we have deleted the sentences”WNK463 and Closantel exhibit stronger anti-tumor activity than VEGFR inhibitor PTK787. This finding is consistent with reports that WNK kinases have other tumor-promoting effects including direct proliferative actions on tumors” in the discussion. We also have edited the conclusion section.

Round 2

Reviewer 3 Report

The authors do not meet my requests:

Qestion 1: "Introduction: This part must be deepened. In particular literature data on the role of WNK1 in cancer should be described"

The added sentences (see below) are not clear

"WNK1 is associated with lncRNA LEF1-AS1 in regulating 55 the hepatocellular carcinoma (HCC) progression [18], viability and migration of HBV associated HCC 56 [19], Moreover, WNK1 phosphosignaling were enriched in breast cancer [20], hyperphosphorylated 57 of WNK1 was observed in retinoblastoma via phosphoproteomics analysis [21], overexpression of 58 WNK1 in cervical carcinoma revealed by meta-transcriptome analysis [22]. Based on the literature 59 evidence, WNK1 plays important role in cancer formation.

Page 4 line114-115. Only one out of two wnk1 knockdown reach statistical significance on tumor cell proliferation. A third wnk1 knockdown should be employed to support the conclusion that "the knockdown of wnk1 gene in zebrafish reduces...tumor cell proliferation".

Ans: Thanks for the comments, both wnk1a and wnk1b knockdown reduces tumor cell proliferation, but due to the big variation in wnk1b knockdown, there is no significance.

Sorry, but No significance means no difference.

Authors compare the effect of Wnk1 and OSR1 inhibitors with VEGFR inhibitor reaching the conclusion that the involvement of Wnk1 in angiogenesis is more important than VEGFR. This is NOT correct. These results can be explained on the basis of a different efficacy of these inhibitors. This part should be changed and discussed also in conclusion section.

Ans: Thanks for the constructive comments, we have deleted the sentences”WNK463 and Closantel exhibit stronger anti-tumor activity than VEGFR inhibitor PTK787. This finding is consistent with reports that WNK kinases have other tumor-promoting effects including direct proliferative actions on tumors” in the discussion. We also have edited the conclusion section.

This part has not been discussed.

Author Response

Question 1: "Introduction: This part must be deepened. In particular literature data on the role of WNK1 in cancer should be described"

The added sentences (see below) are not clear

"WNK1 is associated with lncRNA LEF1-AS1 in regulating 55 the hepatocellular carcinoma (HCC) progression [18], viability and migration of HBV associated HCC 56 [19], Moreover, WNK1 phosphosignaling were enriched in breast cancer [20], hyperphosphorylated 57 of WNK1 was observed in retinoblastoma via phosphoproteomics analysis [21], overexpression of 58 WNK1 in cervical carcinoma revealed by meta-transcriptome analysis [22]. Based on the literature 59 evidence, WNK1 plays important role in cancer formation.

Reply: Thanks to the constructive comments, we have described more of the role of WNK1 in carcinogenesis to the introduction, line 54~61.

WNK1 has been reported to be involved in many aspects of carcinogenesis. For example, WNK1 is required for activation of extracellular signal-regulated kinases 5 (ERK5) by epidermal growth factor (EGF) and Mitogen-activated Protein/ERK Kinase Kinases (MEKK2/3) pathway [17], and Wnt/β-catenin signaling [18]. WNK1 is linked to oncogenic phosphoinositide-3-kinase (PI3K)-AKT pathways, transforming growth factor beta (TGF-β) and Nuclear factor Kappa B (NF-κB) pathways [11]. Together, these reports indicate that WNK1 plays an important role in tumor cell growth and remodeling of extracellular matrix for tumor invasion. However, the role of WNK1 in tumor-induced angiogenesis remain unknown.

Question 2: Page 4 line114-115. Only one out of two wnk1 knockdown reach statistical significance on tumor cell proliferation. A third wnk1 knockdown should be employed to support the conclusion that "the knockdown of wnk1 gene in zebrafish reduces...tumor cell proliferation".

Ans: Thanks for the comments, both wnk1a and wnk1b knockdown reduces tumor cell proliferation, but due to the big variation in wnk1b knockdown, there is no significance.

Sorry, but No significance means no difference.

Reply: Thanks to the comments. Zebrafish often have duplicate copies of a gene that is present as a single copy in human orthologue. In this case, human WNK1 orthologue has duplicated copies in zebrafish: wnk1a and wnk1b. In this study, we found wnk1a has a much more significant effect on tumor induced angiogenesis which is consistent to our previous study in embryonic angiogenesis (Lai et al, 2013 PLoS One). We found the expression levels of wnk1b is much weaker than wnk1a from qPCR and in-situ hybridization. Using morpholino knockdown, we also found the defect in embryonic angiogenesis is more severe in wnk1a morphants than wnk1b morphants.

In the result, line 175~182, we have changed the statement to” In both metrics, wnk1a knockdown reduced proliferation of Hep3B cells relative to controls, however, the effect of wnk1b knockdown versus controls did not reach statistical significance. This phenomena is consistent to our previous study in embryonic angiogenesis [16], whereas wnk1a knockdown has more significant reduction embryonic angiogenesis than wnk1b. Overall, the results indicate that knockdown of wnk1a gene in zebrafish reduces tumor-induced angiogenesis and tumor cell proliferation, and knockdown of wnk1b is less effective in both angiogenesis and tumor cell proliferation reduction.”

Question 3: Authors compare the effect of Wnk1 and OSR1 inhibitors with VEGFR inhibitor reaching the conclusion that the involvement of Wnk1 in angiogenesis is more important than VEGFR. This is NOT correct. These results can be explained on the basis of a different efficacy of these inhibitors. This part should be changed and discussed also in conclusion section.

Ans: Thanks for the constructive comments, we have deleted the sentences ”WNK463 and Closantel exhibit stronger anti-tumor activity than VEGFR inhibitor PTK787. This finding is consistent with reports that WNK kinases have other tumor-promoting effects including direct proliferative actions on tumors” in the discussion. We also have edited the conclusion section.

This part has not been discussed.

Reply: Thanks to the comments. We have changed the result section in line 241~246 “The results indicate that WNK1 pathway inhibitors exhibit stronger antitumor activity than the VEGFR inhibitor PTK787 in the xenotransplantation assay, although this result might be simply due to the different efficacy of these inhibitors, another possibility is that WNK kinases might also promote tumor cell proliferation autonomously within the cancer cells, we will examine the role of WNK kinases using endothelial-specific wnk1 overexpression transgenic fish (section 2.8).”

We also edited the discussion section in line 432~438 “In this study, WNK463 and Closantel exhibit stronger anti-tumor activity than VEGFR inhibitor PTK787. These results could be explained on the basis of a different efficacy of these inhibitors. This finding might also imply that WNK kinases have other tumor-promoting effects including direct proliferative actions on tumors [11,32]. Similar to WNK1 downstream target-SPAK functions both downstream and also upstream of NF-κB signaling [33,34] forming a positive feedback loop to promote tumorigenesis. WNK1 might also play multiple roles in tumorigenesis through cross-talk with multiple signal pathways.”

Also, we discussed multiple pathways activated by WNK1 might play roles in tumorigenesis as in line 446~449 “WNK1 is a positive regulator of canonical Wnt/β-catenin signaling in cancer cell line [18]. Wnt/β-catenin pathway promotes cell proliferation, cell survival, and expression of angiogenic factor in endothelial cells [36]. Therefore, it is possible that WNK1 may affect tumor cell proliferation and angiogenesis by activating Wnt/β-catenin signaling.”

Round 3

Reviewer 3 Report

Accept